# The mitotic spindle is chiral due to torques within microtubule bundles

Maja Novak[1,2], Bruno Polak[2], Juraj Simunić [2], Zvonimir Boban[1], Barbara Kuzmić[2], Andreas W. Thomae[3], Iva M. Tolić [2] & Nenad Pavin [1]

Mitosis relies on forces generated in the spindle, a micro-machine composed of microtubules and associated proteins. Forces are required for the congression of chromosomes to the metaphase plate and their separation in anaphase. However, besides forces, torques may exist in the spindle, yet they have not been investigated. Here we show that the spindle is chiral. Chirality is evident from the finding that microtubule bundles in human spindles follow a left-handed helical path, which cannot be explained by forces but rather by torques. Kinesin-5 (Kif11/Eg5) inactivation abolishes spindle chirality. Our theoretical model predicts that bending and twisting moments may generate curved shapes of bundles. We found that bundles turn by about $-2\,\mathrm{deg\,\mu m^{-1}}$ around the spindle axis, which we explain by a twisting moment of roughly $-10\,\mathrm{pN\mu m}$. We conclude that torques, in addition to forces, exist in the spindle and determine its chiral architecture.

[1] Department of Physics, Faculty of Science, University of Zagreb, Bijenička cesta 32, 10000 Zagreb, Croatia. [2] Division of Molecular Biology, Ruđer Bošković Institute, Bijenička cesta 54, 10000 Zagreb, Croatia. [3] Walter Brendel Centre of Experimental Medicine and Core Facility Bioimaging at the Biomedical Center, University of Munich, 82152 Planegg-Martinsried, Germany. These authors contributed equally: Maja Novak, Bruno Polak, Juraj Simunić, Zvonimir Boban, Barbara Kuzmić. Correspondence and requests for materials should be addressed to I.M.Tć. (email: tolic@irb.hr) or to N.P. (email: npavin@phy.hr)

During mitosis, the genetic material is divided into two equal parts by the spindle, a complex and dynamic structure made of microtubules, motor proteins, and other microtubule-associated proteins[1,2]. Microtubules arranged in parallel bundles known as kinetochore fibers extend from the poles and attach to chromosomes via kinetochores, which are protein complexes assembled on the centromeres of each chromosome[3]. Kinetochore fibers exert forces that position the kinetochores in the equatorial plane of the spindle in metaphase[4,5], and pull on kinetochores to separate the chromosomes into the future daughter cells in anaphase[6]. Some of the microtubules that are not associated with kinetochores meet in the central part of the spindle to form antiparallel bundles known as interpolar or overlap bundles[7]. These bundles act as a bridge between sister kinetochore fibers and balance the forces at kinetochores in metaphase and anaphase[8–12], and regulate pole separation in anaphase[13].

Movement of kinetochores typically follows the contour of the attached microtubule bundles. Thus, forces acting on kinetochores have been explored theoretically in one-dimensional models[14,15]. An early model described the interactions between the kinetochore and microtubules by "kinetochore sleeves" to explain the origin of the forces that move chromosomes[16]. Motor proteins, as well as microtubule dynamics, were included in models that explained chromosome movements during metaphase and anaphase[10,17]. Such one-dimensional models were successful in identification of the most important physical mechanisms of chromosome movements in mitosis.

The models that go beyond one dimension were successful in describing forces that generate spindle shape. These models explained the curved shape of spindles with centrosomes by taking into account that microtubules get curved when compressive forces act on them[8,18], as discussed in ref [19]. The curved shape of spindles without centrosomes was explained by considering local interactions of short microtubules in a liquid crystal model[20].

Forces in the spindle are mainly generated by motor proteins[21]. However, in vitro studies have shown that, in addition to forces, several spindle motor proteins including kinesin-5 (Eg5), kinesin-8 (Kip3), kinesin-14 (Ncd), and dynein can generate torque by switching microtubule protofilaments with a bias in a certain direction[22–25]. Thus, torques may exist in the spindle and control the shape and spatial arrangement of microtubule bundles. Yet, torques in the spindle have not been studied so far.

Here we show that the mitotic spindle is a chiral object. We found that microtubule bundles twist around the spindle axis following a left-handed helical path. Inactivation of kinesin-5 (Kif11/Eg5) abolishes the chirality of the spindle, suggesting that this motor has a role in the maintenance of the helical shape of microtubule bundles. We introduce a theoretical model, which predicts that curved shapes cannot be explained by forces but rather by torques. Our quantitative approach allows us to estimate the magnitude of the torques. Our experiments and theory suggest that, in addition to forces, torques exist in the spindle and regulate its chiral architecture.

## Results

**The mitotic spindle is a chiral object with left-handed helicity of microtubule bundles**. We set out to infer forces and torques in the spindle, by using the shape of microtubule bundles. We first used stimulated emission depletion (STED) super-resolution microscopy[26,27] to determine the shapes of microtubule bundles in metaphase spindles in human HeLa and U2OS cells (Fig. 1a and Supplementary Fig. 1a). Single optical sections of spindles showed that microtubule bundles are continuous almost from

pole to pole and acquire complex curved shapes (Fig. 1a). While the outer bundles have a shape resembling the letter C, bundles that look like the letter S are found in the inner part of the spindle. Overall, the majority of the bundles throughout the spindle have contours that fall between these two shapes. Thus, STED images of the spindle suggest that microtubules are arranged into bundles exhibiting a variety of shapes, which run almost through the whole spindle.

In order to obtain three-dimensional (3D) contours of microtubule bundles, we used vertically oriented spindles, which are found occasionally in a population of mitotic cells, and imaged them by confocal microscopy (Fig. 1b, c). In these spindles, optical sections are roughly perpendicular to the bundles, allowing for precise determination of the bundle position in each section and thus of the whole contour (see Methods). We used fixed HeLa cells expressing green fluorescent protein (GFP)-tagged protein regulator of cytokinesis 1 (PRC1) (refs. [28,29]) because it shows the position of overlap bundles and indirectly the position of the coupled kinetochore fibers[8,9], without interference of the signal from polar and astral microtubules. PRC1-labeled bundles, which appear as spots in a single image plane of a vertically oriented spindle, were tracked through the z-stacks (Fig. 1c; see Methods). When imaged in this manner and viewed end-on along the spindle axis, the bundles that have a planar shape would form an aster-like arrangement. Surprisingly, we found that the arrows connecting bottom and top end of each bundle rotate clockwise, implying that bundles follow a left-handed helical path along the spindle axis (Fig. 1c and Supplementary Fig. 1b; Supplementary Movies 1–3). The helicity of bundles, defined as the average change in angle with height (Fig. 1d), where negative numbers denote left-handed helicity, was $-2.5 \pm 0.2 \deg \mu m^{-1}$ (mean ± s.e.m., $n = 415$ bundles from 10 cells). We conclude that the mitotic spindle is a chiral object with left-handed helicity of the microtubule bundles.

To explore the chirality of horizontally oriented spindles, we imaged them and rearranged the z-stacks to obtain the slices perpendicular to the spindle axis, similar to the z-stacks of vertical spindles (Fig. 1e, f; see Methods). Bundles in horizontal spindles showed left-handed helicity as in vertical spindles (Fig. 1f, Supplementary Fig. 1b; Supplementary Movie 4). We noted that horizontal spindles had higher left-handed helicity ($-3.3 \pm 0.2$ deg $\mu m^{-1}$, mean ± s.e.m., $n = 388$ bundles from 10 cells) than vertical ones ($p$ value from a Student's $t$-test = 0.012 (two-tailed and two-sample unequal-variance); for technical controls see Supplementary Fig. 1f, g). Furthermore, we investigated the chirality of spindles in several other conditions: (i) unlabeled HeLa cells with horizontal spindles immunostained for PRC1 (Fig. 1g; Supplementary Movie 5), (ii) and (iii) live HeLa cells expressing PRC1-GFP, with horizontal (Supplementary Fig. 1c; Supplementary Movie 6) and vertical spindles, (iv) live U2OS cells with vertical spindles, expressing mCherry-α-tubulin (Supplementary Movie 7), and (v) unlabeled U2OS cells with horizontal spindles immunostained for PRC1 (Supplementary Fig. 1c; Supplementary Movie 8). In each of these cell populations, we found that microtubule bundles follow a left-handed helical path (Fig. 1h; Supplementary Fig. 1d, e). Taken together, our results suggest that even though helicities vary among different conditions, labeling, spindle orientations, and cell lines, the bundles consistently twist in a left-handed direction with an average helicity of about $-2$ deg $\mu m^{-1}$ (Fig. 1h). We conclude that left-handed chirality is a robust feature of the spindle in the examined cell lines (Fig. 1i).

**Inactivation of kinesin-5 (Kif11/Eg5) reduces spindle chirality, whereas depolymerization of cortical actin does not**. We set out to investigate the mechanical basis of spindle chirality by studying

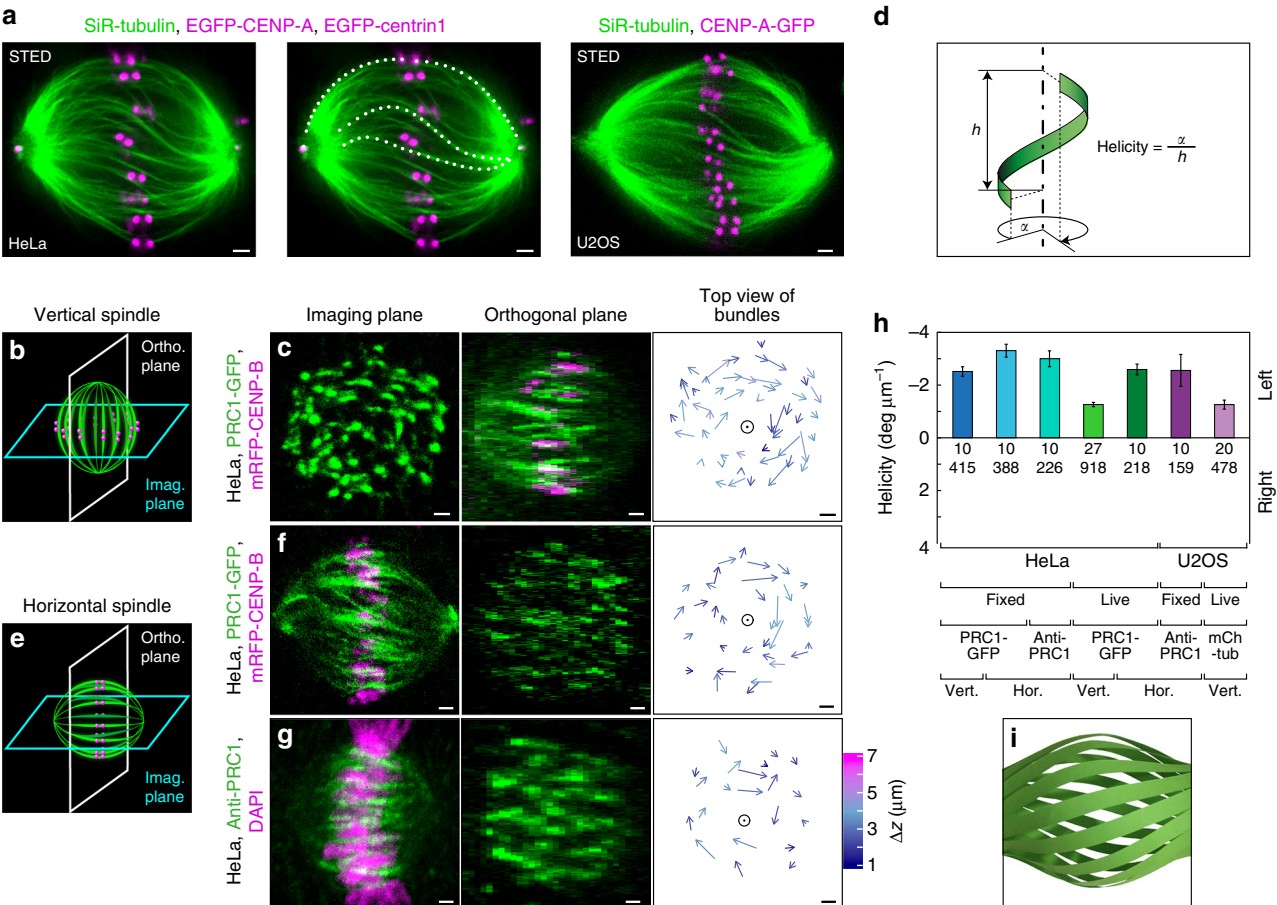

**Fig. 1** Mitotic spindle is chiral. **a** STED image (single *z*-plane) of metaphase spindle in a live HeLa cell expressing EGFP-CENP-A and EGFP-centrin1 (both shown in magenta) (left and middle; middle panel shows traces of microtubule bundles superimposed on the image), and in a live U2OS cell expressing CENP-A-GFP (magenta) (right). Microtubules are labeled with SiR-tubulin (green). **b** Imaging scheme of a vertically oriented spindle. **c** Imaging plane of a vertical spindle in a fixed HeLa cell expressing PRC1-GFP and mRFP-CENP-B (only PRC1-GFP is shown) (left); orthogonal plane of the same spindle (middle); arrows connecting starting and ending points of PRC1-GFP bundles traced upwards (right). Longer arrows roughly correspond to larger twist around the spindle axis (circle), colors show *z*-distance between starting and ending points, see color bar in **g**. **d** Schematic representation of the microtubule bundle helicity measurement. **e** Imaging scheme of a horizontally oriented spindle. **f** Horizontal spindle in a fixed HeLa cell expressing PRC1-GFP and mRFP-CENP-B, legend as in **c**. **g** Horizontal spindle in a fixed unlabeled HeLa cell immunostained for PRC1, with DNA stained by DAPI, legend as in **c**. Images in **c** left, and **f**, **g** middle are single planes; images in **c** middle, and **f**, **g** left are maximum intensity projections of five central planes. **h** Spindle helicity averaged over bundles for different conditions (vertical and horizontal spindles, fixed and live cells) and cell lines as indicated. Cell lines used were: HeLa cells expressing PRC1-GFP (1st, 2nd, 4th, and 5th bars), unlabeled HeLa cells immunostained for PRC1 (3rd bar), unlabeled U2OS cells immunostained for PRC1 (6th bar), U2OS cells expressing CENP-A-GFP, mCherry-α-tubulin, and photoactivatable (PA)-GFP-tubulin (7th bar). Data are representative of 4 independent experiments for unlabeled HeLa and U2OS cells immunostained for PRC1 and 3 independent experiments for all other conditions. Numbers represent the number of cells (top) and bundles (bottom). Data for individual cells are shown in Supplementary Fig. 1e. **i** Paper model of the spindle showing left-handed helicity of microtubule bundles and chirality of the whole spindle. Scale bars, 1 μm; error bars, s.e.m

the contribution of forces generated within microtubule bundles in the central spindle and those exerted by astral microtubules. We first hypothesized that twist is generated within the bundles, by motor proteins that rotate the microtubule while walking, such as kinesin-5 (Kif11/Eg5) (ref. [23]). Kinesin-5 inactivation with *S*-trityl-L-cysteine (STLC) (see Methods)[30,31] caused the bundle traces to change from a clockwise rotation to a more random distribution (Fig. 2a; Supplementary Movie 6). STLC treatment reduced the left-handed helicity threefold in horizontal spindles (Fig. 2b; Supplementary Fig. 2a). Likewise, the average helicities in vertical spindles were close to zero 5 and 10 min after STLC treatment (Fig. 2c; Supplementary Fig. 2a–c), whereas mock treatment did not extensively change the helicity (Fig. 2d; Supplementary Fig. 2a, c). STLC treatment did not change spindle length and width (Supplementary Fig. 2d). In U2OS cells, STLC treatment also abolished spindle chirality (Fig. 2e). Based on these

results, we conclude that kinesin-5 is important for maintenance of spindle chirality.

Twist in the spindle may also be regulated by astral microtubules. To explore this possibility, we treated the cells with latrunculin A (see Methods), an agent that depolymerizes actin cortex[32], thereby disrupting astral microtubule cortical pulling[33–35]. We found no significant change in helicity in latrunculin-treated cells (Fig. 2f, g), which indicates that pulling forces generated by astral microtubules at the cell cortex have a minor effect on the shape of microtubule bundles in the spindle. Taken together, these perturbation experiments suggest that spindle chirality relies mainly on forces generated within microtubule bundles in the central spindle rather than at the cell cortex.

**Theory for shapes of microtubule bundles**. To explore how the observed shapes can be explained from a mechanical perspective, we

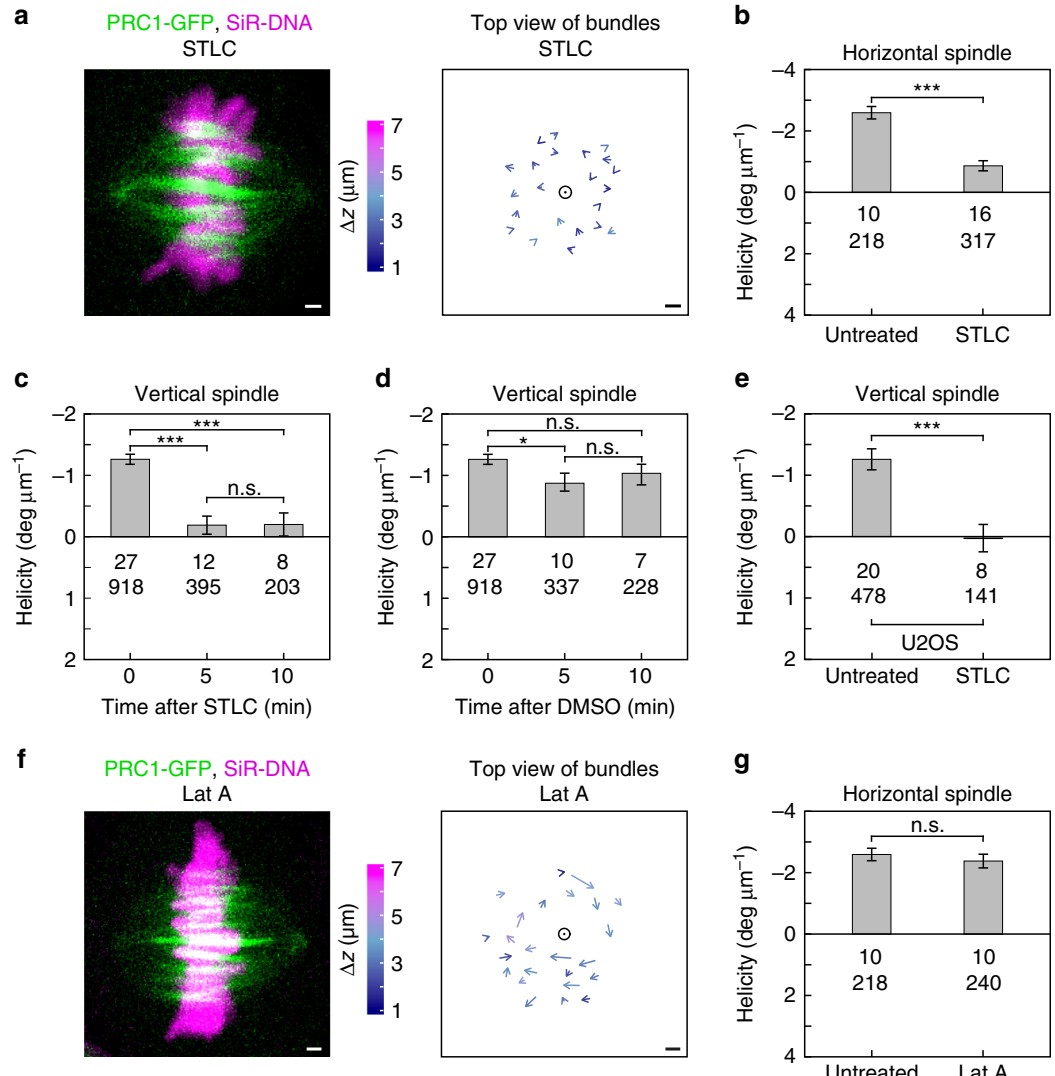

**Fig. 2** Kif11/Eg5 inactivation by STLC reduces spindle chirality, whereas latrunculin A treatment does not. **a** Horizontal spindle in a live HeLa cell expressing PRC1-GFP with SiR-DNA-labeled chromosomes, treated with STLC (left); arrows connecting starting and ending points of bundles traced upwards, from the same cell (right). Circle denotes spindle axis, and colors show z-distance between starting and ending points, see color bar. **b–e** Helicity of spindles before and after STLC or DMSO (mock) treatment, as indicated. **c** Helicity before treatment was different from zero ($p = 10^{-44}$), but not at 5 and 10 min ($p = 0.21$ and 0.28). **d** All helicities were different from zero ($p = 10^{-44}$, $7 \times 10^{-9}$, and $4 \times 10^{-9}$ at 0, 5, and 10 min). **f** Spindle of a live HeLa cell treated with latrunculin A, legend as in **a**. **g** Helicity before and after treatment with latrunculin A. In all panels live HeLa cells expressing PRC1-GFP were used, except in **e** where live U2OS cells expressing CENP-A-GFP and mCherry-α-tubulin were used. In **b-e** and **g**, numbers represent the number of cells (top) and bundles (bottom), from 5 independent experiments in **b-e** and from 4 independent experiments in **g**; ***$p < 0.001$, *$0.01 < p < 0.05$, n.s., not significant; all $p$ values from a Student's $t$-test (two-tailed and two-sample unequal-variance) are given in Supplementary Fig. 2a. Images are maximum intensity projections of five central planes. Scale bars, 1 μm; error bars, s.e.m

introduce a simple physical model of the spindle. The central idea of our theoretical approach is that torques exist within microtubule bundles and generate their helical shapes (Fig. 3a). We describe a microtubule bundle as a thin elastic rod extending between the two spindle poles[18,36] (Fig. 3b), based on our super-resolution images (Fig. 1a). This description is a simplification of the model with three linked rods from ref [8]. The spindle poles are spheres, representing centrosomes together with an adjacent region where most of microtubule bundles are linked together. Based on the observation that the shape of a microtubule bundle can be considered constant during metaphase, in comparison to a quick change of shape after laser cutting[8], we infer that the total forces and torques in an intact bundle are larger than those inducing fluctuations of its shape. Thus, we model a static shape of the spindle, which we describe by a balance of forces and torques at each spindle pole (Fig. 3c) and each

bundle. By taking into account these forces and torques, as well as the elastic properties of microtubule bundles, we calculate the shape of each bundle (Fig. 3d).

**Balance of forces and torques in the spindle and the associated bundle shapes.** In our model, two spindle poles are represented as spheres of radius $d$ with centers separated by vector $\mathbf{L}$ of length $L = |\mathbf{L}|$, and microtubule bundles are represented as curved lines connecting these spheres (Fig. 3b). Microtubule bundles, denoted by index $i = 1, …, n$, extend between points located at the surface of the left and right sphere, where positions with respect to the center of each sphere are given by vectors $\mathbf{d}_i$ and $\mathbf{d}_i'$, respectively. Here, $n$ denotes the number of microtubule bundles. Because the shape of the spindle is static in our model, we introduce a balance

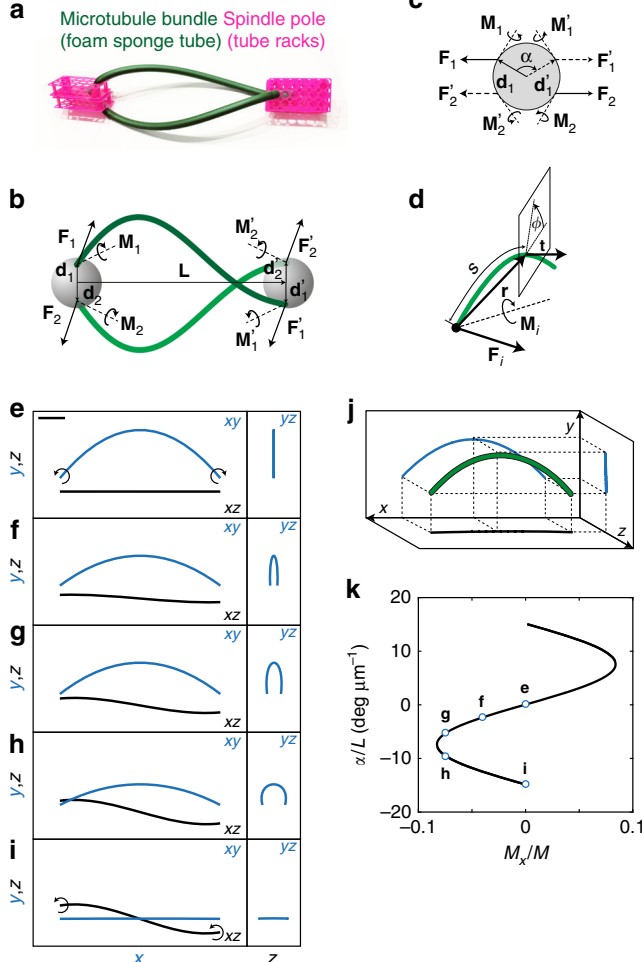

**Fig. 3** Theory for shapes of microtubule bundles. **a** Macroscopic model of the spindle constructed as an illustration of our physical model. **b** Scheme of the model. Microtubule bundles (green) extend between spindle poles (spheres) at the distance **L**. Straight arrows denote forces $\mathbf{F}_{1,2}$, $\mathbf{F}'_{1,2}$ and positions at the spheres $\mathbf{d}_{1,2}$, $\mathbf{d}'_{1,2}$; curved arrows denote torques $\mathbf{M}_{1,2}$, $\mathbf{M}'_{1,2}$. **c** View at the spindle pole along the spindle axis. The angle between the vectors $\mathbf{d}_1$ and $\mathbf{d}'_1$ is $\alpha$; other symbols as in **b**. **d** Scheme of a bundle. Arrows depict contour length $s$, radial vector $\mathbf{r}$, normalized tangent vector $\mathbf{t}$, and torsion angle $\phi$; other symbols as in **b**. **e–i** Predicted shapes of the bundles. Three projections: left, $xy$ (blue), $xz$ (black); right, $yz$ (blue), see scheme in **j**. Parameters are: $\mathbf{M}_1 = (0,0,180)$, $(-5,-30,111)$, $(-10,-70,115)$, $(-10, -128,80)$, $(0,-150,0)$ pNμm for **e–i**, respectively. **j** Scheme of the projections from **e** to **i**. **k** Twist of a microtubule bundle, $\alpha$, divided by spindle length, $L$, as a function of the twisting moment, $M_x = M_{ix}$, normalized to the bending moment, $M = \sqrt{M_{iy}^2 + M_{iz}^2}$ (the same curve for $i = 1,2$). Points denoted by letters e–i correspond to the shapes shown in respective panels. The other parameters are $L = 12\,\mu\text{m}$, $d = 1\,\mu\text{m}$, and $\kappa = 900$ pN μm². Scale bars, 2 μm

of forces and torques for the interaction between the spindle poles and microtubule bundles (Fig. 3c), without describing where the forces and torques are generated. For the left pole, the force balance reads

$$\sum_i \mathbf{F}_i = 0, \tag{1}$$

and the balance of torques reads

$$\sum_i \left(\mathbf{M}_i + \mathbf{d}_i \times \mathbf{F}_i\right) = 0. \tag{2}$$

Here, $\mathbf{F}_i$ and $\mathbf{M}_i$ denote the forces and torques exerted by the left pole at the $i$th microtubule bundle, respectively. They represent a resultant force and torque of all interactions between microtubules and the pole. Balances of forces and torques at the right pole are obtained by replacing $\mathbf{F}_i$, $\mathbf{M}_i$, and $\mathbf{d}_i$ in Eqs. (1) and (2) with $\mathbf{F}'_i$, $\mathbf{M}'_i$, and $\mathbf{d}'_i$, respectively. Here and throughout the text, the prime sign corresponds to the right pole. We also introduce a balance of forces

$$\mathbf{F}_i + \mathbf{F}'_i = 0, \tag{3}$$

and a balance of torques for the microtubule bundle

$$\mathbf{M}_i + \mathbf{M}'_i + \mathbf{d}_i \times \mathbf{F}_i + (\mathbf{L} + \mathbf{d}'_i) \times \mathbf{F}'_i = 0. \tag{4}$$

Forces and torques acting at the microtubule bundle change its shape because microtubule bundles are elastic objects[8,18,36]. We describe a microtubule bundle as a single elastic rod of flexural rigidity $\kappa$ and torsional rigidity $\tau$. The contour of the elastic rod is described by a contour length, $s$, and a vector representing the position in space with respect to the initial point at the sphere representing the spindle pole, $\mathbf{r}(s)$ (Fig. 3d). The normalized tangent vector is calculated as $\mathbf{t} = d\mathbf{r}/ds$. The torsion angle, $\phi(s)$, describes the orientation of the cross-section along the length of the rod. The curvature and the torsion of an elastic rod are described by the static Kirchoff equation[37], which is a generalization of previous models for the curvature of spindle microtubules[8,18]

$$\kappa \mathbf{t} \times \frac{d\mathbf{t}}{ds} + \tau \frac{d\phi}{ds} \mathbf{t} = \mathbf{r} \times \mathbf{F}_i - \mathbf{M}_i. \tag{5}$$

We use this equation to calculate the shapes of microtubule bundles for a set of forces and torques that obey Eqs. (1)–(4).

**Solutions of the model with two bundles.** To investigate the bundle shapes that the model can give, we solve the model as follows. We reduce the complexity of the model by considering a system with two microtubule bundles as a minimal spindle that can attain a curved shape (Fig. 3b; see Methods). Moreover, we impose two symmetries: (i) discrete rotational symmetry of the second order with respect to the major axis, and (ii) symmetry with respect to exchange of the left and right pole (see Methods). Note that mirror symmetries cannot be used due to spindle chirality. In this case, we find that compressive and tensile forces vanish, and thus torques generate curved shapes of the bundles. The analytical solution of the model reads

$$y_i(x) = A_i \left[ \cos\frac{M_{ix}(L-2x)}{2\kappa} \csc\frac{LM_{ix}}{2\kappa} - \cot\frac{LM_{ix}}{2\kappa} \right] - \frac{M_{iy}}{LM_{ix}} x^2 + \frac{M_{iy}}{M_{ix}} x + \frac{LM_{ix}}{2M_{iy}}, \tag{6}$$

$$z_i(x) = A_i \left[ \sin\frac{M_{ix}(L-2x)}{2\kappa} \csc\frac{LM_{ix}}{2\kappa} - 1 \right] + \left( \frac{M_{iz}}{M_{ix}} - \frac{2\kappa M_{iy}}{LM_{ix}^2} \right) x - \frac{LM_{ix}}{2M_{iz}}, \tag{7}$$

with $A_i \equiv \left(-2\kappa M_{iy}M_{iz} - LM_{ix}\left(M_{ix}^2 - M_{iz}^2\right)\right)/2M_{ix}^2 M_{iz}$ and $\left(\frac{2d_i}{M_{ix}L}\right)^2 = \frac{1}{M_{iy}^2} + \frac{1}{M_{iz}^2}$. The derivation of this solution and the solutions for vanishing components of the torque can be found in Methods. Here, free parameters are the twisting and bending

components of the torque, $M_{ix}$, and $M_{iy}$, respectively. The chosen orientation of the coordinate system is such that the solutions are symmetric for $y$ coordinate and antisymmetric for $z$ coordinate (see Methods).

We explore the roles of the twisting and bending components in the generation of shapes and the corresponding helicity (Fig. 3e–j). If torque has a bending component only, we find two solutions which are both planar: the symmetric C-shape if the bending moments at the two ends of the bundle act in the opposite direction (Fig. 3e), and the antisymmetric S-shape if they act in the same direction (Fig. 3i). Interestingly, for torques that include a twisting component, these shapes become 3D and helicity appears (Fig. 3f–h). Individual values of the twisting component result in two different shapes, a deformed C and S (e.g., data points g and h in Fig. 3k, see Supplementary Fig. 3 for solutions with different parameters). Thus, our theory predicts that torques generate curved shapes of bundles, where the twisting component of the torque is required for the helical component of the shape.

We find that the spindle pole size is important for the balance of twisting moments in the spindle, for the following reason. At one spindle pole, two bundles exert the twisting moments in the same direction (Fig. 3b, c). These moments are balanced by the torque at the same pole acting in the opposite direction, arising from response forces exerted by both bundles at this pole (Eqs. (2) and (4), see also Methods). A larger radius of the spindle pole implies a larger lever arm for this force and thus a larger torque (Supplementary Fig. 3).

**Comparison of the model with experiments**. Finally, we compare the results of our model with the experimentally observed shapes of microtubule bundles. We fit our analytical solutions (Eqs. (6) and (7)) to the 3D traces of bundles. With two fitting parameters and a free choice of the orientation of the coordinate system (see Methods), our theory reproduces the whole range of 3D shapes observed in experiments (Fig. 4a). The quality of fits is visible in all three projections of the shapes ($xy$, $xz$, and $yz$ projections in Fig. 4a). These shapes span from simple planar C-shapes without helicity, which are linear in the $yz$ projection, to more complex shapes with different extent of helicity, which are curved in the $yz$ projection.

For 52 bundles the twisting moment was $-8.4 \pm 0.8$ pNμm and the bending moment was $139 \pm 7$ pNμm (data points in Fig. 4b). Our quantification of the torque components is indirect because the values are obtained by fitting the model to the experimentally measured shapes. The negative twisting moment generates the negative helicity observed in the experiments. Theoretical values of helicity increase with increasing twisting moment (curve in Fig. 4b). The fitted data points are found in proximity to the theoretical curve, as expected, if the theory explains well the experiments. In conclusion, our simple model together with experiments suggests that torques, in addition to forces, exist in the spindle and determine its chiral shape.

## Discussion

Chirality is an intriguing property of the biological world, present at all scales ranging from molecules to whole organisms[38]. We found that the human mitotic spindle is a chiral object due to twisting moment within microtubule bundles. This twisting moment results in the rotation of the bundle cross-section along its length, suggesting that individual microtubules within the bundle twist around each other like metal wires in a steel wire rope. Microtubules that twist in such a manner have been observed in yeast spindles[39,40], which consist of a single rod-shaped microtubule bundle. Recently, 3D reconstructions of the

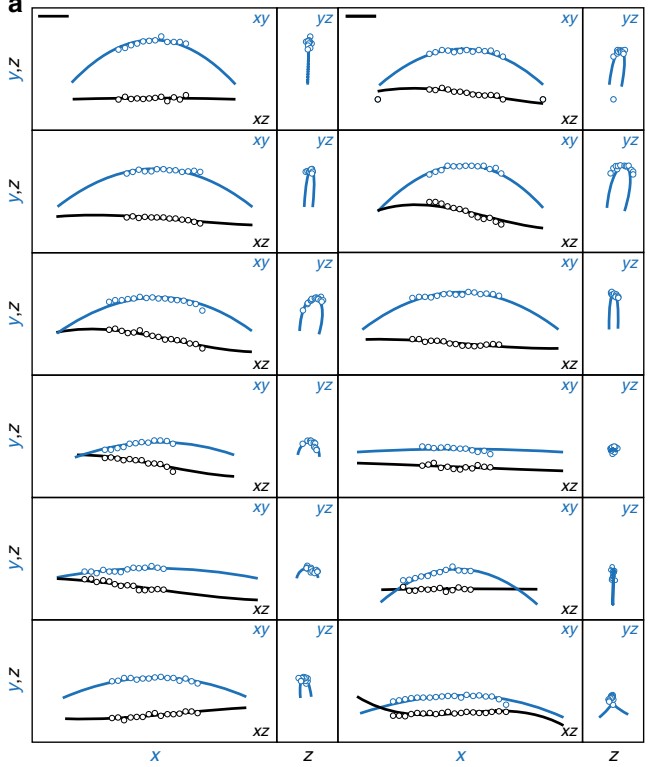

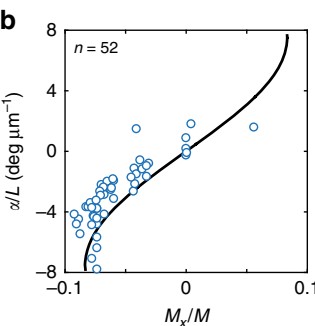

**Fig. 4** Comparison of theory and experiments. (**a**) Theoretical fits (curves) to the traces of microtubule bundles from horizontal spindles in live HeLa cells expressing PRC1-GFP (circles). Three different projections are shown: left, horizontal $xy$ projection (blue), and $xz$ projection (black); right, $yz$ projection (blue). Parameters are, left column: $\mathbf{M} = (0, -5, 180)$, $(-5, -23, 113)$, $(-10, -68, 117)$, $(-8, -55, 69)$, $(-3, -26, 27)$, $(4, 18, 68)$ pNμm, right column: $\mathbf{M} = (-11, -43, 159)$, $(-20, -94, 208)$, $(-5, -31, 108)$, $(0, -2, 8)$, $(1, 0, 154)$, $(-8, 167, 71)$ pNμm. Parameter $L$ is taken from measurements. The other parameters are $d = 1$ μm and $\kappa = 900$ pNμm². (**b**) Theoretical curve representing twist of a microtubule bundle, $\alpha$, divided by spindle length, $L$, as a function of the twisting moment, $M_x = M_{ix}$, normalized to the bending moment, $M = \sqrt{M_{iy}^2 + M_{iz}^2}$ (the same curve for $i = 1, 2$). Circles represent experimental helicity of the traces of microtubule bundles from live HeLa cells expressing PRC1-GFP, as a function of the normalized twisting moment, obtained from fits. The parameters of theoretical curve are $d = 1$ μm, $L = 12$ μm and $\kappa = 900$ pN μm². Scale bars, 2 μm

microtubule organization in the spindles of higher eukaryotic cells have become available[41,42]. By using this approach, it will be interesting to explore the presence of twist in different species and to what extent microtubules within individual bundles twist around each other.

Our experiments showed that kinesin-5 is important for spindle chirality. We speculate that kinesin-5 turns antiparallel microtubules around each other while sliding them apart, which generates torque in the microtubule bundles and consequently their helical shape. Moreover, given that kinesin-5 is localized mainly close to the spindle pole[43], it may have a role in the generation of torque at the pole. Alternatively, linear forces acting on microtubules may lead to torsion due to a helical arrangement of tubulin subunits in the microtubule[44,45]. However, in our experiments with kinesin-5 inactivation spindle length did not change, suggesting that linear forces did not change, thus the observed change in spindle chirality is most likely due to torque exerted by this motor. Finally, in addition to kinesin-5, other mitotic motors, such as kinesin-14, kinesin-8, and dynein[22,24,25], may be involved in the generation of torque. Future studies will reveal the precise molecular mechanism and the contribution of different molecular players to the torque in the spindle and the related chirality.

Our theory together with experiments suggests that the twisting moment in the microtubule bundle is around $-10$ pN$\mu$m and the bending moment 140 pN$\mu$m. Experiments with optical tweezers have shown that single kinesin-1 motors can generate torque up to about 1.65 pN$\mu$m[46]. Assuming that the mitotic motors required for spindle chirality generate a similar amount of torque, we speculate that 10–100 motors per bundle can produce the observed helical shapes of the bundles.

Current models for spindle mechanics describe the collective behavior of motor proteins and how they generate pulling and pushing forces, but not the torque[14–17]. Torques generated by motor proteins have been included in theoretical studies of beat patterns of cilia and flagella[47,48], showing that torques are crucial to explain the helical swimming trajectories of cells such as sperms[49,50]. It will be interesting to develop a model that combines the knowledge about the collective forces of motor proteins in the spindle with the collective torques, to explore the resulting shapes of microtubule bundles, as well as kinetochore movements and the movement of the microtubule lattice towards the spindle pole known as poleward flux[51].

Our work revealed spindle chirality in metaphase, where spindle shape is constant. The theoretical and experimental approaches introduced here could be used to explore the role of torques in the phases of mitosis characterized by spindle shape changes, such as spindle formation in prometaphase[52] and chromosome segregation accompanied with spindle elongation in anaphase[10].

## Methods

**Cell lines**. HeLa-Kyoto BAC lines stably expressing PRC1-GFP were courtesy of Ina Poser and Tony Hyman (Max Planck Institute of Molecular Cell Biology and Genetics, Dresden, Germany). HeLa cells stably expressing EGFP-CENP-A and EGFP-centrin1 were a courtesy of Emanuele Roscioli and Andrew McAinsh (University of Warwick). Human U2OS cells, both unlabeled and permanently transfected with CENP-A-GFP, mCherry-α-tubulin, and photoactivatable (PA)-GFP-tubulin, were courtesy of Marin Barišić and Helder Maiato (University of Porto). Cells were grown in Dulbecco's modified Eagle's medium (DMEM) with Ultraglutamine (Lonza, Basel, Switzerland) supplemented with 10% fetal bovine serum (FBS; Life Technologies, Carlsbad, CA, USA), penicillin, streptomycin, and geneticin (Santa Cruz Biotechnology Inc., Dallas, TX, USA). The cells were kept at 37 °C and 5% $CO_2$ in a Galaxy 170S $CO_2$ humidified incubator (Eppendorf, Hamburg, Germany). All used cell lines were confirmed to be mycoplasma free by using MycoAlert Mycoplasma Detection Kit (Lonza, Basel, Switzerland).

**Sample preparation**. To visualize kinetochores and identify metaphase in experiments on fixed cells, HeLa cells expressing PRC1-GFP cells were transfected by electroporation using Nucleofector Kit R (Lonza, Basel, Switzerland) with the Nucleofector 2b Device (Lonza, Basel, Switzerland), using the high-viability O-005 program. Transfection protocol provided by the manufacturer was followed. Twenty-five to thirty-five hours before imaging, $1 \times 10^6$ cells were transfected with 2.5 μg of monomeric red fluorescent protein (mRFP)-CENP-B plasmid DNA

(pMX234) provided by Linda Wordeman (University of Washington). To visualize chromosomes and determine the metaphase state of the spindle in experiments on live cells, 1 h prior to imaging silicon rhodamine (SiR)-DNA (ref. [53]) (Spirochrome AG, Stein am Rhein, Switzerland) was added to the dish with HeLa cells at a final concentration of 100 nM. For labeling of microtubules with SiR-tubulin[54] (Spirochrome AG, Stein am Rhein, Switzerland), the dye was added to cells at a final concentration of 50–100 nM, 16 h prior to imaging. To prepare samples for microscopy, HeLa and U2OS cells were seeded and cultured in 1.5 ml DMEM medium with supplements at 37 °C and 5% $CO_2$ on uncoated 35-mm glass coverslip dishes, No. 1.5 coverglass (MatTek Corporation, Ashland, MA, USA).

**Drug treatments**. The stock solution of STLC and latrunculin A were prepared in dimethyl sulfoxide (DMSO) to a final concentration of 1 mM. Both drugs and solvent were obtained from Sigma-Aldrich. The working solution was prepared in DMEM at 100 μM. At the time of treatment, the working solution was added to cells at 1:1 volume ratio to obtain a final concentration of 50 μM (the half-maximal inhibitory concentration for STLC in HeLa cells is 700 nM)[30]. Spindles that are already in metaphase when STLC is added retain their shape, whereas spindles that begin to assemble in the presence of the drug become monopolar[55,56]. STLC-treated PRC1-GFP HeLa cells with vertical spindles were imaged as follows: a z-stack of a metaphase spindle before treatment was acquired, then the drug was added and the same spindle was imaged after 5 and 10 min. Appearance of monopolar spindles in the neighborhood of the imaged spindle confirmed the effect of STLC. U2OS cells were treated in the same way, but imaged only after 10 min. For STLC treatment of cells with horizontally oriented spindles in PRC1-GFP HeLa cells, the drug was added at a final concentration of 50 μM and the cells were incubated at 37 °C for 5 min. The cells with metaphase spindles were imaged within 25 min after incubation. For experiments with latrunculin A, PRC1-GFP HeLa cells were treated with 2 μM latrunculin A for 1 h prior to imaging, which was done between 1 and 2 h post treatment. The effect of latrunculin A was confirmed by retraction and rounding of the interphase cells[57]. For mock-treated experiments, cells with vertical spindles were treated with the concentration of DMSO that was used for preparation of the drugs. Vertical spindles that rotated so that the angle between the major axis and z-axis was larger than roughly 30° at 5 or 10 min after treatments were not analyzed.

**Immunostaining**. Unlabeled U2OS and HeLa cells were fixed in ice-cold 100% methanol for 3 min and washed. To permeabilize cell membranes, cells were incubated in Triton (0.5% in phosphate-buffered saline (PBS)) at room temperature for 25 min. To block unspecific binding of antibodies, cells were incubated in 1% normal goat serum (NGS) in PBS for 1 h at 4 °C. Cells were then incubated in 250 μl of primary antibody solution (4 μg ml$^{-1}$ in 1% NGS in PBS) for 48 h at 4 °C. Mouse monoclonal anti-PRC1 antibody (C-1; sc-376983, Santa Cruz Biotechnology, USA) was used. After washing off the primary antibody solution, cells were incubated in 250 μl of secondary antibody solution (4 μg ml$^{-1}$ in 2% NGS in PBS; Alexa Fluor 488 preadsorbed donkey polyclonal anti-mouse IgG, Ab150109; Abcam, Cambridge, UK) for 1 h at room temperature protected from light. After each incubation step, cells were washed three times for 5 min in PBS softly shaken at room temperature. In HeLa cells, we occasionally observed shrinkage of the spindle upon fixation; therefore, for the analysis we only chose spindles which were longer than 9 μm.

**STED microscopy**. STED images of HeLa and U2OS cells were recorded at the Core Facility Bioimaging at the Biomedical Center, LMU Munich. STED resolution images were taken of SiR-tubulin signal, whereas GFP signal of kinetochores and centrin1 was taken at confocal resolution. Gated STED images were acquired with a Leica TCS SP8 STED 3X microscope with pulsed white light laser excitation at 652 nm and pulsed depletion with a 775 nm laser (Leica, Wetzlar, Germany). The objective used was HC PL APO CS2 ×93/1.30 GLYC with a motorized correction collar. Scanning was done bidirectionally at 30–50 Hz, a pinhole setting of 0.93 AU (at 580 nm), and the pixel size was set to $20 \times 20$ nm. The signals were detected with Hybrid detectors with the following spectral settings: SiR-tubulin (excitation 652; emission: 662–715 nm; counting mode; gating: 0.35–6 ns) and GFP (excitation 488; emission 498–550; counting mode, no gating). We estimated that the resolution was roughly 80 nm, based on the measured distance between two centrioles in the same centrosome[58].

In comparison with confocal microscopy, STED microscopy allowed us to better resolve individual bundles in the region close to the spindle pole. However, imaging with STED gives fewer photons because it is done on smaller sample volumes and due to the limitations of labeling with SiR-tubulin dye. High concentrations (higher than 100 nM) of this taxol-based dye occasionally altered spindle appearance, whereas lower concentrations (lower than 50 nM) did not produce enough signal for a super-resolution image. Moreover, imaging of the whole z-stack of the spindle in STED resolution was too slow (5–10 s per imaging plane) to allow for a complete 3D stack to be acquired before the spindle movement compromises the stack acquisition. For reviews discussing STED and other super-resolution microscopy techniques see refs [59–61].

**Confocal microscopy.** Fixed HeLa cells expressing PRC1-GFP were imaged using a Leica TCS SP8 X laser scanning confocal microscope with a HC PL APO ×63/1.4 oil immersion objective (Leica, Wetzlar, Germany). For excitation, a 488-nm line of a visible gas Argon laser and a visible white light laser at 575 nm were used for GFP and mRFP, respectively. GFP and mRFP emissions were detected with HyD (hybrid) detectors in ranges of 498–558 nm and 585–665 nm, respectively. Pinhole diameter was set to 0.8 μm. Images were acquired at 30–60 focal planes with 0.5 μm z-spacing, 30 nm xy-pixel size, and 400 Hz unidirectional xyz scan mode. The system was controlled with the Leica Application Suite X Software (1.8.1.13759, Leica, Wetzlar, Germany). Live HeLa and all U2OS cells were imaged using Bruker Opterra Multipoint Scanning Confocal Microscope[62] (Bruker Nano Surfaces, Middleton, WI, USA). The system was mounted on a Nikon Ti-E inverted microscope equipped with a Nikon CFI Plan Apo VC ×100/1.4 numerical aperture oil objective (Nikon, Tokyo, Japan). During imaging, cells were maintained at 37 °C in Okolab Cage Incubator (Okolab, Pozzuoli, NA, Italy). A 60 μm pinhole aperture was used and the xy-pixel size was 83 nm. For excitation of GFP and mCherry fluorescence, a 488 and a 561 nm diode laser line was used, respectively. The excitation light was separated from the emitted fluorescence by using Opterra Dichroic and Barrier Filter Set 405/488/561/640. Images were captured with an Evolve 512 Delta EMCCD Camera (Photometrics, Tucson, AZ, USA) with no binning performed. To cover the whole metaphase spindle, z-stacks were acquired at 30–60 focal planes separated by 0.5 μm with unidirectional xyz scan mode. The system was controlled with the Prairie View Imaging Software (Bruker Nano Surfaces, Middleton, WI, USA).

**Theory: solution for two bundles and imposed symmetries.** Our model describes a system consisting of $n$ microtubule bundles, where torques and forces can vary between bundles, resulting in a system with a large number of degrees of freedom. To reduce the number of degrees of freedom, we consider a case with two microtubule bundles, $i = 1,2$. Further, we use rotational symmetry of the spindle with respect to the major axis by imposing the symmetry for forces $\mathbf{F}_{1\parallel} = \mathbf{F}_{2\parallel}$, $\mathbf{F}_{1\perp} = -\mathbf{F}_{2\perp}$ and for torques $\mathbf{M}_{1\parallel} = \mathbf{M}_{2\parallel}$, $\mathbf{M}_{1\perp} = -\mathbf{M}_{2\perp}$. Here, index $\parallel$ and $\perp$ denotes components of vectors that are parallel and perpendicular to the vector $\mathbf{L}$, respectively, obeying $\mathbf{F}_i = \mathbf{F}_{i\parallel} + \mathbf{F}_{i\perp}$ and $\mathbf{M}_i = \mathbf{M}_{i\parallel} + \mathbf{M}_{i\perp}$. In addition, we impose that the magnitude of torque is equal at both poles, $|\mathbf{M}_i| = |\mathbf{M}_i'|$, that the components of torque parallel to $\mathbf{L}$ are balanced, $\mathbf{M}_{i\parallel} = -\mathbf{M}_{i\parallel}'$, and $\mathbf{d}_i \cdot \mathbf{M}_{i\perp} = \mathbf{d}_i' \cdot \mathbf{M}_{i\perp}' = 0$. For simplicity, we also choose that vectors $\mathbf{d}_i$ and $\mathbf{d}_i'$ are perpendicular to $\mathbf{L}$.

To solve the model, we choose a Cartesian coordinate system such that $x$-axis is parallel to $\mathbf{L}$ and $\mathbf{d}_2 = -\mathbf{d}_1$. In this coordinate system, radial vector has components $\mathbf{r} = (x, y, z)$ and torques have components $\mathbf{M}_i = (M_{ix}, M_{iy}, M_{iz})$. The orientation of the coordinate system is chosen such that $M_{iy} = M_{iy}'$, whereas the $z$-component obeys $M_{iz} = -M_{iz}'$, giving

$$\mathbf{M}_i' = (-M_{ix}, M_{iy}, -M_{iz}). \qquad (8)$$

From Eq. (1) and the symmetry $\mathbf{F}_{1\parallel} = \mathbf{F}_{2\parallel}$, we obtain that $F_{1x} = F_{2x} = 0$. From the symmetry $\mathbf{F}_{1\perp} = -\mathbf{F}_{2\perp}$, the other two components obey $F_{1y} = -F_{2y}$, $F_{1z} = -F_{2z}$. By taking the symmetries into account, the $z$-component of Eq. (4) reads $LF_{1y} = 0$, and consequently the $y$-component of the force vanishes, $F_{iy} = 0$. By using the $x$-component of Eq. (2), which reads $M_{ix} + d_{iy}F_{iz} = 0$, we calculate the $z$-component of the force, giving

$$\mathbf{F}_i = (0, 0, -M_{ix}/d_{iy}). \qquad (9)$$

By combining this equation and the relationship obtained from the $y$-component of Eq. (4), which reads $2M_{iy} + LF_{iz} = 0$, we obtain

$$d_{iy} = M_{ix}L/2M_{iy}. \qquad (10)$$

By using the $x$-component of Eq. (4), together with Eqs. (3), (8), and (9), we obtain $d_{iy} = d_{iy}'$. Because we imposed the symmetry $\mathbf{d}_i \cdot \mathbf{M}_{i\perp} = \mathbf{d}_i' \cdot \mathbf{M}_{i\perp}' = 0$, and $\mathbf{d}_i$ is perpendicular to $\mathbf{L}$, the $z$-components of vectors $\mathbf{d}_i$ and $\mathbf{d}_i'$ obey

$$d_{iz} = -d_{iy}M_{iy}/M_{iz} \qquad (11)$$

and $d_{iz} = -d_{iz}'$, respectively. By using $d_{iy}^2 + d_{iz}^2 = d^2$, we obtain the relation between the parameters $M_{ix}$, $M_{iy}$ and $M_{iz}$, $\left(\frac{2d_i}{M_{ix}L}\right)^2 = \frac{1}{M_{iy}^2} + \frac{1}{M_{iz}^2}$. Thus, our model has three free parameters, $M_{ix}$, $M_{iy}$, and the choice of the coordinate system orientation described above.

**Analytical solutions.** We solve Eq. (5) by using a Cartesian coordinate system in which this equation is given by a system of three nonlinear differential equations. In a small angle approximation, where $ds \approx dx$, these equations simplify and

become linear:

$$-\tau \frac{d\phi}{dx} = M_{ix}, \qquad (12)$$

$$-\kappa \frac{d^2z}{dx^2} - M_{ix}\frac{dy}{dx} = F_{ix}z - F_{iz}x - M_{iy}, \qquad (13)$$

$$\kappa \frac{d^2y}{dx^2} - M_{ix}\frac{dz}{dx} = -F_{ix}y + F_{iy}x - M_{iz}, \qquad (14)$$

The estimate of the error due to this approximation is given in the subsection "Fitting of the model to experimentally observed shapes." Note that, in this approximation, the torsional rigidity $\tau$ affects the orientation of the cross-section of the microtubule bundle (Eq. (12)), whereas it does not affect the 3D contour of the cross-section center explicitly (Eqs. (13) and (14)). Because we do not study the orientation of the cross-section of the microtubule bundle, the torsional rigidity does not appear in the analytical solutions used for fitting the experimental data (Eqs. (6) and (7)).

Analytical solutions of Eqs. (13) and (14), together with $F_{ix} = 0$, read:

$$y_i(x) = A_i \sin(\omega_i x + \phi_i) + \frac{F_{iz}x^2}{2M_{ix}} + \left(\frac{M_{iy}}{M_{ix}} + \frac{\kappa F_{iy}}{M_{ix}^2}\right)x + B_i, \qquad (15)$$

$$z_i(x) = A_i \cos(\omega_i x + \phi_i) - \frac{F_{iy}x^2}{2M_{ix}} + \left(\frac{M_{iz}}{M_{ix}} + \frac{\kappa F_{iz}}{M_{ix}^2}\right)x + C_i, \qquad (16)$$

where $\omega_i = M_{ix}/\kappa$. Integration constants $A_i$, $B_i$, $C_i$, $\phi_i$ are obtained from the boundary conditions $y_i(0) = y_i(L) = d_{iy}$, $z_i(0) = -z_i(L) = d_{iz}$, where $d_{iy}$ and $d_{iz}$ are given by Eqs. (10) and (11). The final expressions are given in the main text in Eqs. (6) and (7). In the special case of vanishing twisting moment, $M_{ix} = 0$, Eqs. (6) and (7) reduce to: (i) $y(x) = [2d\kappa + M_{1z}(L-x)x]/2\kappa$, $z(x) = 0$ in the case with non-vanishing $M_{1z}$ and (ii) $y(x) = 0$, $z(x) = (L-2x)[6d\kappa + M_{1y}x(-L + x)]/6\kappa L$ in the case with non-vanishing $M_{1y}$.

**Choice of parameter values.** The size of the spindle pole, representing centrosomes together with an adjacent region where most of microtubule bundles are linked together, is estimated to be $d = 1$ μm. The distance between the spindle poles, $L$, is obtained from the experimental measurements.

The flexural rigidity of the microtubule bundle is calculated as $\kappa = N_{MT}\kappa_0 = 900$ pNμm², where $N_{MT} = 30$ is the number of microtubules in the bundle[63,64] and $\kappa_0 = 30$ pNμm² is the flexural rigidity of a single microtubule[36]. Here we use the assumption that the microtubules in a bundle are allowed to slide with respect to each other when the bundle deforms, as in our previous work[8]. However, if microtubules are cross-linked in a manner that does not allow for sliding, then the flexural rigidities would scale as the microtubule number squared[65].

**Fitting of the model to experimentally observed shapes.** We have compared the theoretically obtained shapes, given by Eqs. (13) and (14), to the tracking data of horizontal spindles from live HeLa cells expressing PRC1-GFP. The parameters of the fit are $M_{1x}$ and $M_{1y}$, together with the orientation of the coordinate system of the tracked shape. Used parameters are $d = 1$ μm and $\kappa = 900$ pNμm². Parameter $L$ is obtained from the experimentally measured distance between the poles. We fitted 61 traced bundles, and for 52 of all the shapes discrepancy between fitted curves and experimental data was:

$$\frac{\sum_j (y(X_j) - Y_j)^2}{N} + \frac{\sum_j (z(X_j) - Z_j)^2}{N} < 0.1.$$ Here, $X_j$, $Y_j$, $Z_j$ are measured coordinates of the tracked point $j$, and $N$ denotes number of data points used for fitting a single bundle. For the fitting, we used only bundles with maximal distance from the major axis larger than 1 μm, with minimum 12 tracked points and minimum 3 tracked points on each side of the spindle equatorial plane.

The small angle approximation used for the fitting overestimates the bending and twisting moments by 10% for typical bundles in the spindle, whereas for the outermost bundles the error is up to 30%. The estimate of the error on the bending moment is based on the comparison of $M_z$ from the model with the exact solution for the shape of an elastic rod with the torque $\mathbf{M} = (0, 0, M_z)$ and vanishing forces, which is a circle of a radius $R = \kappa/M_z$. We used the arc of the circle that reproduces the bundle shape obtained by the model to estimate the radius $R$ and to calculate $M_z$ from the exact solution. The estimate of the error on the twisting moment is equal to that of the bending moment because the exact solution for the helicity is a function of the ratio of the twisting moment to the bending moment (Fig. 4b).

**Image analysis.** Microscopy images were analyzed in Fiji Software[66]. For the analysis of horizontal spindles, only the spindles with both poles roughly in the same plane were used to ensure that spindles are maximally vertical after the transformation into vertical orientation, which was done by using a code written in R programming language[67] in RStudio. Before the transformation, the z-stack of

the spindle in a single channel was rotated in Fiji so that the spindle major axis was approximately parallel to the $x$-axis. Signal intensity at each pixel in a z-stack is denoted as $I(i, j, k)$, where indices $i, j$ denote coordinates in the imaging plane, and $k$ denotes the number of the imaging plane of the z-stack. To transform the 3D image of the spindle into vertical orientation, we applied the transformation $I'(i, j, k) = I(k, i, j)$, which preserves the orientation (handedness) of the coordinate system, that is, corresponds to rotation of the image without mirroring. The coordinates $(i, j, k)$ correspond to 3D positions $(x, y, z) = (i \cdot$ pixel size, $j \cdot$ pixel size, $k \cdot z$-distance). The aberrations caused by refractive index mismatch between immersion oil and aqueous sample were taken into account by multiplying z-step size by a correction factor of 0.81 to obtain the correct z-distance. We calculated this factor as a ratio of the cell diameter in $y$ and $z$ direction, assuming that a mitotic cell is spherical[68] (Supplementary Fig. 1f). This value is consistent with theoretical predictions for z-aberrations due to refractive index mismatch[69] and experimental measurements[70].

Bundles in 3D images of spindles oriented vertically (including transformed images of horizontal spindles and images of vertical spindles) were tracked manually using Multipoint tool in Fiji (Supplementary Movies 2 and 7). Individual bundles were determined by moving through the z-stack. Because microtubule bundles appear as spots in a single z-image, each point was placed at the center of the signal. Moving up and down through the z-stack helped to determine this point. Each bundle was tracked through all z-planes where it appears as a single spot. In addition, positions of the spindle poles were determined as the focus point where the PRC1 signal on the microtubule bundles, which is faint in the region close to the pole, ends (Supplementary Movies 2 and 7). Coordinates of bundles and poles from images of vertical spindles were transformed so that both poles are on the z-axis. For the analysis of helicity only the tracked points in the central part of the spindle, between 0.3 and 0.7 of the normalized spindle length, were taken into account. We used only bundles with average distance from the major axis larger than 1.35 μm.

**Statistical analysis**. Graphs were generated in the programming language R. Fiji was used to scale images and adjust brightness and contrast. Figures were assembled in Adobe Illustrator CS5 and Adobe Photoshop CS5 (Adobe Systems, Mountain View, CA, USA). Data are given as mean ± s.e.m., unless otherwise stated. Significance of data was estimated by Student's $t$-test (two-tailed and two-sample unequal-variance). $p < 0.05$ was considered statistically significant. Values of all significant differences are given with degree of significance indicated (*$0.01 < p < 0.05$, **$0.001 < p < 0.01$, ***$p < 0.001$). The number of analyzed cells and microtubule bundles is given in the respective figure panel.

**Code availability**. The code used in this study is available from the corresponding author upon reasonable request.

## Data availability
The authors declare that all data supporting the findings of this study are available within the article and its supplementary information files. The coordinates of the tracked microtubule bundles from all cells used for the analysis are deposited to figshare (https://doi.org/10.6084/m9.figshare.6736997).

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

## Acknowledgements

We thank Andrew McAinsh, Emanuele Roscioli, Ina Poser, Tony Hyman, Marin Barišić, and Helder Maiato for cell lines; Igor Weber, Maja Marinović, Vedrana Filić Mileta,and the rest of the Weber lab for help with the confocal microscope. We thank Steffen Dietzel, Anna H. Klemm, and the Core Facility Bioimaging at the Biomedical Center—LMU, Munich, Germany for help with STED microscopy. We also thank Ivana Šarić for the drawings. We express our gratitude to Vukušić, all other members of Tolić and Pavin groups, and Stephan Grill for discussions. This work was funded by the European Research Council (ERC Consolidator Grant, GA number 647077, granted to I.M.T.), Unity through Knowledge Fund (UKF, project 18/15, granted to N.P. and I.M.T.), and the European Social Fund (HR.3.2.01-0022, co-leader I.M.T.). We also acknowledge support from the QuantiXLie Center of Excellence, a project cofinanced by the Croatian Government and European Union through the European Regional Development Fund—the Competitiveness and Cohesion Operational Programme (Grant KK.01.1.1.01.0004, element leader N.P.), and the Croatian Science Foundation (HRZZ, project IP-2014-09-4753, granted to I.M.T.).

## Author contributions

M.N. developed the theoretical model. J.S., B.P., and B.K. performed confocal microscopy experiments. Z.B. together with B.K., J.S., and B.P. analyzed the experimental data. J.S. and B.K. carried out STED imaging, with A.T. providing expertise on STED microscopy. N.P. and I.M.T. conceived the project and supervised theory and experiments, respectively. I.M.T., N.P., J.S., and M.N. wrote the paper with input from all authors.

## Additional information

**Competing interests:** The authors declare no competing interests.

