## [Peer Review File · Nature Communications]

Reviewers' comments:

Reviewer #1 (Remarks to the Author):

In this manuscript, the authors observed metaphase spindles of human cultured cells by super-resolution fluorescence microscopy (STED) and found that the spindle is chiral with helical paths of the microtubule bundles. Interestingly, this chirality is lost by inhibiting kinesin-5, a key microtubule motor for formation of the bipolar spindle. They created a theoretical model based on the physical properties of the microtubule bundles (bending elasticity, torsional rigidity) and the two spindle poles that exert forces and torques at the end of the bundle, and demonstrated that the helical twist of microtubule bundles observed in the cells can be reproduced by adjusting the parameters.

Even though the building blocks are chiral, a larger structure made of their assembly might not necessarily be chiral at the lower spatial resolution. In this sense, the chirality of the mitotic spindle is not trivial but potentially interesting from the point of views of biophysics and structural biology. However, arguments about its significance in normal cells without any experimental markers are insufficient. In their theoretical approach, the rationale for assuming that the torques are applied only at the ends of microtubule on the surface of the spindle pole is unclear. Furthermore, they haven't paid enough attention to the physiological relevances of the parameters used for fitting their model with the experimental data and of the resulting estimations of the torques. Thus, the current manuscript is not suitable for publication as it is.

Major points

1. Existence of the spindle chirality in normal cells

To assess the spindle structure, the authors mainly observed live cells treated with SiR-tubulin or expressing PRC1-GFP. A concern is that both of these markers might affect the dynamics of microtubules and thus the chirality of the spindle might have been exaggerated. SiR-tubulin is a fluorescent derivative of docetaxel, a taxane, which binds and stabilizes microtubules. PRC1 is a potent microtubule-bundling and stabilizing protein and just a slight over-expression can cause abnormal interphase perinuclear ring-like bundles (Mollinari, C., et al. (2002) J Cell Biol 157, 1175–1186.). A BAC PRC1-GFP cell line provided by Hyman lab shows such interphase bundles in a minor population of the cells (our unpublished observation), indicating that the level of total PRC1 is higher than usual in this cell line although it is unclear whether it is exactly the same line as the one used in this study. Twist of microtubules could be introduced by abnormal elongation of spindle microtubules in a limited space, or by imbalanced polymerization/depolymerization dynamics between kinetochore microtubules and interpolar microtubules. On the other hand, the consistent left-handedness and the sensitivity to kinesin-5 inhibition suggest that the spindle chirality is not just an artifact.

It is important to confirm that the chirality is observed in normal cells without any markers that might affect microtubule dynamics. Among various combinations tested by the authors, "U2OS anti-PRC1, fixed" in Figure 1d and Supplementary Figure 1c is the most important. The final helicity of this class of $\sim -2.5^\circ/\mu\text{m}$ is consistent with the other measurements. However, I couldn't find any images or movies of them. It would be ideal to show images or movies of this category and fixed normal HeLa cells stained with an anti-tubulin antibody as well (strong twisting as in Figure 1a is not very common).

By the way, the descriptions of the images are sometimes confusing. For example, in Figure 1, it is not clear whether the images shown are derived from live recordings or from the fixed cells. Supplementary Video 5 has two colors, but there is no description about what are shown in which color channel. For each image/movie, information about cell type, markers used, and live/fixed should be clearly indicated.

There have been published 3D reconstitutions of the microtubule organization in the spindles of

mammalian cells by electron microscopy. The most recent example is Nixon, F.M., et al. (2017) *J Cell Sci* 130, 1845–1855. The microtubules in these reconstructions don't seem to be twisted as reported in this manuscript. This is also the case for the cryo-EM reconstructed metaphase spindles in *C. elegans* embryos (Redemann, S., et al. (2017) *Nat Commun* 8, 15288.). Some discussion should be necessary.

2.Theoretical model

It is not clear whether the assumption of torques at the end of microtubule bundles is reasonable as a model for the twisting of microtubule bundles sensitive to the inhibition of kinesin-5, which is a linear motor that moves along the lateral surface of a microtubule. If the spindle chirality and its sensitivity to STLC are both true, then an essential question would be how kinesin-5, a microtubule bundling linear motor, can twist microtubule bundles.

Formula 5 contains tau, a parameter for torsional rigidity. However, the solutions, formula 6 and 7 don't contain it. It is not clear how tau disappeared. It is counter-intuitive that the shape of the twist is independent of the torsional rigidity of the microtubule bundles.

For the determination of moments by fitting, kappa was set to $\kappa=900 \text{ pN}\mu\text{m}^2$. What is the rationale for this? Are the resulting values of the twisting moment $-8 \text{ pN}\mu\text{m}$ and bending moment $140 \text{ pN}\mu\text{m}$ reasonable? Can kinesin-5 generate this twisting moment? Is it consistent with the actual measurement of the torque component generated by kinesin-5/Eg5 (ref 21)?

Reviewer #2 (Remarks to the Author):

Novak and co-workers reported about the study of the mitotic spindle, a complex and dynamic structure made of microtubules, motor proteins and other microtubule-associated proteins which divides the genetic materials in two identical part during the mitosis. They developed a physical model to explain the shapes of the microtubule bundles in the spindle. Central part and novelty in the model are that torques exist within microtubule bundles and generate their helical shapes. Their model found good agreement with experimental data obtained applying a pool of high-resolution fluorescence microscopy technique, such as single-point confocal, multi-point confocal and STED microscopy.

The work is well described, data and conclusions are robust. The entire work is an excellent example of synergy between theoretical modeling and experimental validation; thereby I am supportive of publication in *Nature Communications* after a minor revision which takes into account my suggestions listed below.

Major comment:

Novak and co-workers used a pool of different microscopy techniques, which provide diffraction-unlimited resolution or diffraction-limited resolution. However, it is not clear to me which motivations drove the authors to use a technique rather another. I image that STED microscopy has been used for its superior resolution ability and multi-point confocal for the excellent temporal resolution. However, will be nice that the authors better explain this in the text. For readers that would like to take advantages from this work for their applications (as the authors claim in the Discussion Section) would be important to find responses to technical questions, such as: the resolution of confocal microscopy is not enough for this studies? Why is STED microscopy not used across the whole studies? Why is live-cell imaging performed only with confocal microscopy? Can three-dimensional imaging be performed with STED microscopy as well?

Minor comments:

- I suggest authors include in the references some recent reviews about STED microscopy. Seminal publications of STED microscopy are important, but in this context, where readers may be more interested in the application of STED, reviews are also important.
- Figure 1a. It is not clear to me what magenta denotes in the left and center images. Do they

represent the same image? The center image is different only for the traces? Both EGFP-CENP-A and EGFP-Cnetrin1 are expressed in the images? Please clarify this

Reviewer #3 (Remarks to the Author):

In a well-written paper, with ample supporting experimental data, the authors claim:

- (1) the mitotic spindle exhibits a chiral structure;
- (2) the chiral structure is due to torques within the bundle, tentatively identified as being exerted by kinesin motors within the spindle.

The experimental evidence for spiral structure in the spindle is convincing. Though spindle chirality has not been extensively studied in the literature, the same conclusion was arrived at and presented in the same Am. Soc. Cell Biol. meeting in which the authors also first reported their results (Y.

u, Redemann, et al., 2017).

The second major claim is that intrinsic torques within the spindle give rise to the chiral structure. However, in this case, both the experimental evidence and the theoretical model are not as strong, and open to several questions.

The most fundamental (unstated) assumption is that torque is required in order to observe torsion in a microtubule or kinetochore. Such an assumption is common in standard mechanical beam theory (e.g., Euler-Bernoulli, Timoshenko beam models), and results from the uncoupling of fundamental vibrational modes for a homogeneous, isotropic beam undergoing small deformation. However, the cross sectional structure of a microtubule or kinetochore is neither homogeneous or isotropic. Furthermore, the strains exhibited by mitotic spindle microtubules are relatively large, beyond the range of applicability of linear beam theories. In such cases all standard deformation modes (compression, bending, twist) are typically coupled and torsion can be obtained from compressional forces only, without any torque. This has been well documented in standard mechanical engineering beam theory (e.g., composite beam structures from the aerospace literature), and recent theoretical work has also established this for microtubules. It is not at all clear that the linear model used in this paper (Eq 5) is actually a valid representation.

On the experimental side, while kinesin-8 has been associated with a left-stepping bias that might impart torque as referenced in the paper, this reviewer is not aware of comparable evidence for kinesin-5, inactivated here by STLC. The loss of chirality might be simply due to inactivation of longitudinal forces exerted by kinesin-5, that due to the structure of the microtubule can lead to torsion even when no torque is directly exerted by kinesin-5.

The theoretical fit of the linear beam model to observed shapes is (Fig. 4) is good supporting evidence for the authors' claim, but not fully convincing. Solutions to the linear beam model are similar to the standard fourth-order Euler-Bernoulli beam model. Two boundary conditions are fixed, the centrosome attachment points. Then only two parameters would be needed to characterize the curves, hence it is not surprising that the observed geometrical shapes are well captured with only two parameters as the authors state (Mix, Miy). Note that a 0.3 2-norm error (Methods, p. 18, authors give 2-norm squared error of 0.1) is not a particularly accurate fit. As an aside, the value for the torsional rigidity (τ) does not seem to be mentioned in the paper.

In conclusion, the paper finds previously unreported chiral structure in the mitotic spindle, concurrently reported by another research group. The theoretical motivation for the observed structure (inherent torques in the kinetochores) might be valid, but this is not clear at present since alternative mechanisms have not been considered.

Rebuttal letter

Reviewers' comments:

Reviewer #1 (Remarks to the Author):

In this manuscript, the authors observed metaphase spindles of human cultured cells by super-resolution fluorescence microscopy (STED) and found that the spindle is chiral with helical paths of the microtubule bundles. Interestingly, this chirality is lost by inhibiting kinesin-5, a key microtubule motor for formation of the bipolar spindle. They created a theoretical model based on the physical properties of the microtubule bundles (bending elasticity, torsional rigidity) and the two spindle poles that exert forces and torques at the end of the bundle, and demonstrated that the helical twist of microtubule bundles observed in the cells can be reproduced by adjusting the parameters.

Even though the building blocks are chiral, a larger structure made of their assembly might not necessarily be chiral at the lower spatial resolution. In this sense, the chirality of the mitotic spindle is not trivial but potentially interesting from the point of views of biophysics and structural biology. However, arguments about its significance in normal cells without any experimental markers are insufficient. In their theoretical approach, the rationale for assuming that the torques are applied only at the ends of microtubule on the surface of the spindle pole is unclear. Furthermore, they haven't paid enough attention to the physiological relevances of the parameters used for fitting their model with the experimental data and of the resulting estimations of the torques. Thus, the current manuscript is not suitable for publication as it is.

Major points

1. Existence of the spindle chirality in normal cells

To assess the spindle structure, the authors mainly observed live cells treated with SiR-tubulin or expressing PRC1-GFP. A concern is that both of these markers might affect the dynamics of microtubules and thus the chirality of the spindle might have been exaggerated. SiR-tubulin is a fluorescent derivative of docetaxel, a taxane, which binds and stabilizes microtubules. PRC1 is a potent microtubule-bundling and stabilizing protein and just a slight over-expression can cause abnormal interphase perinuclear ring-like bundles (Mollinari, C., et al. (2002) *J Cell Biol* 157, 1175–1186.). A BAC PRC1-GFP cell line provided by Hyman lab shows such interphase bundles in a minor population of the cells (our unpublished observation), indicating that the level of total PRC1 is higher than usual in this cell line although it is unclear whether it is exactly the same line as the one used in this study. Twist of microtubules could be introduced by abnormal elongation of spindle microtubules in a limited space, or by imbalanced polymerization/depolymerization dynamics between kinetochore microtubules and interpolar microtubules. On the other hand, the consistent left-handedness and the sensitivity to kinesin-5 inhibition suggest that the spindle chirality is not just an artifact.

It is important to confirm that the chirality is observed in normal cells without any markers that might affect microtubule dynamics. Among various combinations tested by

the authors, “U2OS anti-PRC1, fixed” in Figure 1d and Supplementary Figure 1c is the most important. The final helicity of this class of $\sim -2.5^\circ/\mu\text{m}$ is consistent with the other measurements. However, I couldn’t find any images or movies of them. It would be ideal to show images or movies of this category and fixed normal HeLa cells stained with an anti-tubulin antibody as well (strong twisting as in Figure 1a is not very common).

Authors: We thank the reviewer for these suggestions. We performed new experiments, in which we measured twist in fixed unlabeled HeLa cells stained with an anti-PRC1 antibody (Fig. 1g, h Supplementary Movie 5). The average helicity in these cells was consistent with that from other cell lines (Fig. 1h). In the revised manuscript, we also show images and a movie of U2OS anti-PRC1, fixed (Supplementary Fig. 1c Supplementary Movie 8). We have revised the text on p. 5 to include these new data:

“Furthermore, we investigated the chirality of spindles in several other conditions: (i) unlabeled HeLa cells with horizontal spindles immunostained for PRC1 (Fig. 1g; Supplementary Movie 5), (ii) and (iii) live HeLa cells expressing PRC1-GFP, with horizontal (Supplementary Fig. 1c; Supplementary Movie 6) and vertical spindles, (iv) live U2OS cells with vertical spindles, expressing mCherry- α -tubulin (Supplementary Movie 7), and (v) unlabeled U2OS cells with horizontal spindles immunostained for PRC1 (Supplementary Fig. 1c; Supplementary Movie 8). In each of these cell populations, we found that microtubule bundles follow a left-handed helical path (Fig. 1h, Supplementary Fig. 1d, e).”

Reviewer #1: By the way, the descriptions of the images are sometimes confusing. For example, in Figure 1, it is not clear whether the images shown are derived from live recordings or from the fixed cells. Supplementary Video 5 has two colors, but there is no description about what are shown in which color channel. For each image/movie, information about cell type, markers used, and live/fixed should be clearly indicated.

Authors: We have added the missing information about the cell type, markers, and live or fixed condition of the cell in all figure and movie captions.

Reviewer #1: There have been published 3D reconstitutions of the microtubule organization in the spindles of mammalian cells by electron microscopy. The most recent example is Nixon, F.M., et al. (2017) J Cell Sci 130, 1845–1855. The microtubules in these reconstructions don’t seem to be twisted as reported in this manuscript. This is also the case for the cryo-EM reconstructed metaphase spindles in *C. elegans* embryos (Redemann, S., et al. (2017) Nat Commun 8, 15288.). Some discussion should be necessary.

Authors: To discuss these papers, we added the following text in Discussion: “Recently, 3D reconstructions of the microtubule organization in the spindles of higher eukaryotic cells have become available^{41, 42}. By using this approach, it will be interesting to explore the presence of twist in different species and to what extent microtubules within individual bundles twist around each other.”

Reviewer #1:

2.Theoretical model

It is not clear whether the assumption of torques at the end of microtubule bundles is reasonable as a model for the twisting of microtubule bundles sensitive to the inhibition of kinesin-5, which is a linear motor that moves along the lateral surface of a microtubule. If the spindle chirality and its sensitivity to STLC are both true, then an essential question would be how kinesin-5, a microtubule bundling linear motor, can twist microtubule bundles.

Authors: In our model, torques exist within bundles and not only at their ends (p. 6). To further emphasize this point, we extended the description on p. 7: "... They represent a resultant force-torque of all interactions between microtubules and the pole, without describing where the forces and torques are generated."

To discuss how kinesin-5 can twist microtubule bundles, we extended the following paragraph in Discussion:

"Our experiments showed that kinesin-5 is important for spindle chirality. We speculate that kinesin-5 turns antiparallel microtubules around each other while sliding them apart, which generates torque in the microtubule bundles and consequently their helical shape. Moreover, given that kinesin-5 is localized mainly close to the spindle pole⁴³, it may have a role in the generation of torque at the pole. Alternatively, linear forces acting on microtubules may lead to torsion due to a helical arrangement of tubulin subunits in the microtubule⁴⁴. However, in our experiments with kinesin-5 inactivation spindle length did not change, suggesting that linear forces did not change, thus the observed change in spindle chirality is most likely due to torque exerted by this motor. Finally, in addition to kinesin-5, other mitotic motors, such as kinesin-14, kinesin-8, and dynein^{22, 24, 25} may be involved in the generation of torque. Future studies will reveal the precise molecular mechanism and the contribution of different molecular players to the torque in the spindle and the related chirality."

Reviewer #1: Formula 5 contains tau, a parameter for torsional rigidity. However, the solutions, formula 6 and 7 don't contain it. It is not clear how tau disappeared. It is counter-intuitive that the shape of the twist is independent of the torsional rigidity of the microtubule bundles.

Authors: The shapes of the microtubule bundles are indeed independent of the torsional rigidity and we agree that this is counterintuitive. To clarify this point, we added the following text in Methods:

"Note that, in this approximation, the torsional rigidity \$\tau\$ affects the orientation of the cross-section of the microtubule bundle (equation (12)), whereas it does not affect the 3D contour of the cross-section center explicitly (equations (13) and (14)). Because we do not study the orientation of the cross-section of the microtubule bundle, the torsional rigidity does not appear in the analytical solutions used for fitting the experimental data (equations (6) and (7))."

Reviewer #1: For the determination of moments by fitting, kappa was set to kappa=900 pN μ m². What is the rationale for this? Are the resulting values of the twisting moment –

8 pN μ m and bending moment 140 pN μ m reasonable? Can kinesin-5 generate this twisting moment? Is it consistent with the actual measurement of the torque component generated by kinesin-5/Eg5 (ref 21)?

Authors: To better explain the rationale for parameter values including kappa, we added the following text in Methods:

“Choice of parameter values. The size of the spindle pole, representing centrosomes together with an adjacent region where most of microtubule bundles are linked together, is estimated to be $d = 1 \mu\text{m}$. The distance between the spindle poles, L , is obtained from the experimental measurements.

The flexural rigidity of the microtubule bundle is calculated as $\kappa = N_{MT}\kappa_0 = 900 \text{ pN}\mu\text{m}^2$, where $N_{MT} = 30$ is the number of microtubules in the bundle^{63, 64} and $\kappa_0 = 30 \text{ pN}\mu\text{m}^2$ is the flexural rigidity of a single microtubule³⁶. Here we use the assumption that the microtubules in a bundle are allowed to slide with respect to each other when the bundle deforms, as in our previous work⁸. However, if microtubules are cross-linked in a manner that does not allow for sliding, then the flexural rigidities would scale as the microtubule number squared⁶⁵.

”

To discuss the resulting values of torque and compare them with the torque generated by motors (note that torque has not been measured for kinesin-5, but only for kinesin-1), we added the following text in Discussion:

“Our theory together with experiments suggests that the twisting moment in the microtubule bundle is around -10 pN μ m and the bending moment 140 pN μ m. Experiments with optical tweezers have shown that single kinesin-1 motors can generate torque up to about 1.65 pN μ m⁴⁵. Assuming that the mitotic motors required for spindle chirality generate a similar amount of torque, we speculate that 10-100 motors per bundle can produce the observed helical shapes of the bundles.”

Reviewer #2 (Remarks to the Author):

Novak and co-workers reported about the study of the mitotic spindle, a complex and dynamic structure made of microtubules, motor proteins and other microtubule-associated proteins which divides the genetic materials in two identical part during the mitosis. They developed a physical model to explain the shapes of the microtubule bundles in the spindle. Central part and novelty in the model are that torques exist within microtubule bundles and generate their helical shapes. Their model found good agreement with experimental data obtained applying a pool of high-resolution fluorescence microscopy technique, such as single-point confocal, multi-point confocal and STED microscopy.

The work is well described, data and conclusions are robust. The entire work is an excellent example of synergy between theoretical modeling and experimental validation;

thereby I am supportive of publication in Nature Communications after a minor revision which takes into account my suggestions listed below.

Major comment:

Novak and co-workers used a pool of different microscopy techniques, which provide diffraction-unlimited resolution or diffraction-limited resolution. However, it is not clear to me which motivations drove the authors to use a technique rather another. I image that STED microscopy has been used for its superior resolution ability and multi-point confocal for the excellent temporal resolution. However, will be nice that the authors better explain this in the text. For readers that would like to take advantages from this work for their applications (as the authors claim in the Discussion Section) would be important to find responses to technical questions, such as: the resolution of confocal microscopy is not enough for this studies? Why is STED microscopy not used across the whole studies? Why is live-cell imaging performed only with confocal microscopy? Can three-dimensional imaging be performed with STED microscopy as well?

Authors: To address the technical questions about microscopy, we added the following paragraph in Material and Methods:

“In comparison with confocal microscopy, STED microscopy allowed us to better resolve individual bundles in the region close to the spindle pole. However, imaging with STED gives fewer photons because it is done on smaller sample volumes and due to the limitations of labeling with SiR-tubulin dye. High concentrations (higher than 100 nM) of this taxol-based dye occasionally altered spindle appearance, whereas lower concentrations (lower than 50 nM) did not produce enough signal for a superresolution image. Moreover, imaging of the whole z-stack of the spindle in STED resolution was too slow (5-10 seconds per imaging plane) to allow for a complete 3D stack to be acquired before the spindle movement compromises the stack acquisition. For reviews discussing STED and other superresolution microscopy techniques see Refs. ⁵⁹⁻⁶¹.”

Reviewer #2: Minor comments:

- I suggest authors include in the references some recent reviews about STED microscopy. Seminal publications of STED microscopy are important, but in this context, where readers may be more interested in the application of STED, reviews are also important.

Authors: We added review papers on STED, see our response to the previous comment.

Reviewer #2: - Figure 1a. It is not clear to me what magenta denotes in the left and center images. Do they represent the same image? The center image is different only for the traces? Both EGFP-CENP-A and EGFP-Cnetrin1 are expressed in the images? Please clarify this

Authors: We revised the figure caption to clarify these issues:

"STED image of a metaphase spindle in a **live** HeLa cell expressing EGFP-CENP-A and EGFP-centrin1 (**both shown in magenta**) (**left and middle; middle panel shows traces of microtubule bundles superimposed on the image**), and in a **live** U2OS cell expressing CENP-A-GFP (**magenta**) (**right**)."

Reviewer #3 (Remarks to the Author):

In a well-written paper. with ample supporting experimental data, the authors claim:

- (1) the mitotic spindle exhibits a chiral structure;
- (2) the chiral structure is due to torques within the bundle, tentatively identified as being exerted by kinesin motors within the spindle.

The experimental evidence for spiral structure in the spindle is convincing. Though spindle chirality has not been extensively studied in the literature, the same conclusion was arrived at and presented in the same Am. Soc. Cell Biol. meeting in which the authors also first reported their results (Yu, Redemann, et al., 2017).

Authors: We checked all abstracts from the Am. Soc. Cell Biol. Meeting 2017. There is only one abstract by Yu and Redemann (copied below), in which spindle chirality is not mentioned.

P2096

Board Number: B237

Microtubules push chromosomes apart in anaphase.

C. Yu¹, S. Redemann², H. Wu³, T.Y. Yeon¹, T. Müller-Reichert², D.J. Needleman^{1,4}; □
¹School of Engineering and Applied Sciences, Harvard University, Cambridge, MA, ²Medical Faculty Carl Gustav Carus, Technische Universität Dresden, Dresden, Germany, ³Department of Physics, Harvard University, Cambridge, MA, ⁴Department of Molecular and Cellular Biology and Center for Systems Biology, Harvard University, Cambridge, MA

*The spindle partitions chromosomes into the two daughter cells during cell division. The manner by which the spindle generates and exerts the forces that segregate chromosomes in anaphase remain poorly understood. We used genetic perturbations, electron tomography, laser ablation, and quantitative optical microscopy to study chromosome segregation in *Caenorhabditis elegans* mitotic and meiotic spindles, and human mitotic spindles. All of these spindles contain a population of microtubules between chromosomes in anaphase. Perturbing these inter-chromosomal microtubules greatly influences chromosome motion, while interfering with microtubules between chromosomes and spindle poles has a minimal impact. In addition, these inter-chromosomal microtubules slide apart in the same speed as sister chromosomes separate, but in a faster speed than the spindle poles move apart, suggesting that their sliding is coupled to chromosome motion. Our results argue that inter-chromosomal microtubules push chromosomes apart by a combination of polymerization and sliding. Based on these findings, we propose that pushing from inter-chromosomal microtubules is the primary driver of chromosome motion in anaphase.*

Reviewer #3: The second major claim is that intrinsic torques within the spindle give rise to the chiral structure. However, in this case, both the experimental evidence and the theoretical model are not as strong, and open to several questions.

The most fundamental (unstated) assumption is that torque is required in order to observe torsion in a microtubule or kinetochore. Such an assumption is common in standard mechanical beam theory (e.g., Euler-Bernoulli, Timoshenko beam models), and results from the uncoupling of fundamental vibrational modes for a homogeneous, isotropic beam undergoing small deformation. However, the cross sectional structure of a microtubule or kinetochore is neither homogeneous or isotropic. Furthermore, the strains exhibited by mitotic spindle microtubules are relatively large, beyond the range of applicability of linear beam theories. In such cases all standard deformation modes (compression, bending, twist) are typically coupled and torsion can be obtained from compressional forces only, without any torque. This has been well documented in standard mechanical engineering beam theory (e.g., composite beam structures from the aerospace literature), and recent theoretical work has also established this for microtubules. It is not at all clear that the linear model used in this paper (Eq 5) is actually a valid representation.

Authors: We use the linear model (Eq. 5), because the strain in the spindle microtubules, defined as the ratio of the cross-section diameter and the bending radius, is small (up to 0.005). This estimate is based on the radius of curvature of 5 μm or larger, obtained from the fits to the experimental measurements (Fig. 4), which is at least 200 times larger than the diameter of the microtubule (25 nm).

In the revised manuscript, we expanded the motivation for Eq. 5 by citing previous linear models for the curvature of spindle microtubules:

“The curvature and the torsion of an elastic rod are described by the static Kirchoff equation³⁷, which is a generalization of previous models for the curvature of spindle microtubules^{8, 18},”

Reviewer #3: On the experimental side, while kinesin-8 has been associated with a left-stepping bias that might impart torque as referenced in the paper, this reviewer is not aware of comparable evidence for kinesin-5, inactivated here by STLC. The loss of chirality might be simply due to inactivation of longitudinal forces exerted by kinesin-5, that due to the structure of the microtubule can lead to torsion even when no torque is directly exerted by kinesin-5.

Authors: In vitro experiments have provided evidence that kinesin-5 generates torque on the microtubule (Ref. 21 in the original manuscript). To make this point clearer, we have expanded the text in Introduction to list all the motors for which similar behavior has been observed:

“However, *in vitro* studies have shown that, in addition to forces, several spindle motor proteins including kinesin-5 (Eg5), kinesin-8 (Kip3), kinesin-14 (Ncd), and dynein can generate torque by switching microtubule filaments with a bias in a certain direction²²⁻²⁵.”

To address the possibility of inactivation of longitudinal forces exerted by kinesin-5, we performed additional measurements of spindle length and width (Supplementary Fig. 2d) in control cells and after inactivation of kinesin-5. We found that kinesin-5 inactivation did not change spindle length and width, which is now reported in Results section (p. 5) and discussed in Discussion:

“Alternatively, linear forces acting on microtubules may lead to torsion due to a helical arrangement of tubulin subunits in the microtubule⁴. However, in our experiments with kinesin-5 inactivation spindle length did not change, suggesting that linear forces did not change, thus the observed change in spindle chirality is most likely due to torque exerted by this motor.”

Reviewer #3: The theoretical fit of the linear beam model to observed shapes is (Fig. 4) is good supporting evidence for the authors' claim, but not fully convincing. Solutions to the linear beam model are similar to the standard fourth-order Euler-Bernoulli beam model. Two boundary conditions are fixed, the centrosome attachment points. Then only two parameters would be needed to characterize the curves, hence it is not surprising that the observed geometrical shapes are well captured with only two parameters as the authors state (Mix, Miy). Note that a 0.3 2-norm error (Methods, p. 18, authors give 2-norm squared error of 0.1) is not a particularly accurate fit. As an aside, the value for the torsional rigidity (τ) does not seem to be mentioned in the paper.

Authors: We agree that it is not surprising that the observed geometrical shapes are well captured with only two parameters. We deleted the word “only” in the revised manuscript. We also agree that certain fits are not particularly accurate. This is due to the fact that we use a simple model, which does not include variability among different microtubule bundles in the same spindle. Regarding the torsional rigidity, we added the following text in Methods:

“Note that, in this approximation, the torsional rigidity τ affects the orientation of the cross-section of the microtubule bundle (equation (12)), whereas it does not affect the 3D contour of the cross-section center explicitly (equations (13) and (14)). Because we do not study the orientation of the cross-section of the microtubule bundle, the torsional rigidity does not appear in the analytical solutions used for fitting the experimental data (equations (6) and (7)).”

Reviewer #3: In conclusion, the paper finds previously unreported chiral structure in the mitotic spindle, concurrently reported by another research group. The theoretical motivation for the observed structure (inherent torques in the kinetochores) might be valid, but this is not clear at present since alternative mechanisms have not been considered.

Authors: In the revised manuscript, we address alternative mechanisms in Discussion (p. 10), as explained two points above.

REVIEWERS' COMMENTS:

Reviewer #1 (Remarks to the Author):

The authors have addressed most of my points by adding measurements (Fig. 1) and providing better explanations of their theory. One point that remains not very obvious is whether the assumption $ds \approx dx$, which is essential for the analytical solution, is reasonable (line 446). s is a function of x and depends on the shape of the microtubule bundles (the local helicity and the distance from the pole-to-pole axis), which is indeed the solution of the equation 5. Further reasoning with the experimental measurements and consistency with the mathematical solution would be helpful.

(minor points) The descriptions of the lengths of the scale bars are missing in Fig. 1, Fig. 3 and Fig. 4.

Reviewer #3 (Remarks to the Author):

Reviewer #3 thanks authors for their response and manuscript revisions. My previous reference to concurrent work by Yu, Redemann et al was mistaken due to a search engine error.

The only point remaining is attribution of torsion to molecular motor torque. The mention in the authors' revision of alternative mechanisms is sufficient at this point to proceed with publication and allow further discussion in the wider scientific community since the experimental evidence in the paper should be disseminated. I however do not agree with the arguments put forward in the authors' Rebuttal letter.

(1) Even within a linear model coupling between extension and torsion is quite possible, possibly strong, and well documented:

- "The torsion-extension coupling in pretwisted elastic beams", S. Krenk, International Journal of Solids and Structures

Volume 19, Issue 1, 1983, Pages 67-72

- "On Timoshenko-like modeling of initially curved and twisted composite beams", By: Yu, WB (Yu, WB); Hodges, DH (Hodges, DH); Volovoi, V (Volovoi, V); Cesnik, CES (Cesnik, CES)

INTERNATIONAL JOURNAL OF SOLIDS AND STRUCTURES

Volume: 39

Issue: 19

Pages: 5101-5121

Published: SEP 2002

- "Direct determination of DNA twist-stretch coupling"

By: Kamien, RD (Kamien, RD); Lubensky, TC (Lubensky, TC); Nelson, P (Nelson, P); O'Hern, CS (O'Hern, CS)

EUROPHYSICS LETTERS

Volume: 38

Issue: 3

Pages: 237-242

Published: APR 20 1997

The key hypothesis of a Kirchhoff rod that is not satisfied for a microtubule is that the cross-section of a segment remains undistorted and perpendicular to axis defined by third dimension (e.g., Cytoskeleton, 75(2):45-60, 2018). Eq. (5) is then no longer valid since longitudinal force directly produces torsion. The argument that inactivation of kinesin-5 did not lead to overall change of spindle length is an interesting conjecture, but not convincing since they are not the sole determinants of spindle length. The interesting prediction from the paper is the number of

motors (10-100) per bundle needed to produce the twisting moment. This is worth further experimental study.

Given the above, the claim in the Abstract that "bending and twisting moments generate curved shapes" seems too strong, and should be replaced by "may generate", similar to the more careful statement of conclusions in the revised Discussion section.

Rebuttal letter

Reviewers' comments:

Reviewer #1 (Remarks to the Author):

The authors have addressed most of my points by adding measurements (Fig. 1) and providing better explanations of their theory. One point that remains not very obvious is whether the assumption $ds \approx dx$, which is essential for the analytical solution, is reasonable (line 446). s is a function of x and depends on the shape of the microtubule bundles (the local helicity and the distance from the pole-to-pole axis), which is indeed the solution of the equation 5. Further reasoning with the experimental measurements and consistency with the mathematical solution would be helpful.

Authors: To clarify this issue we added the following paragraph in Methods:

“The small angle approximation used for the fitting overestimates the bending and twisting moments by 10% for typical bundles in the spindle, whereas for the outermost bundles the error is up to 30%. The estimate of the error on the bending moment is based on the comparison of M_z from the model with the exact solution for the shape of an elastic rod with the torque $\mathbf{M} = (0,0,M_z)$ and vanishing forces, which is a circle of a radius $R = \kappa/M_z$. We used the arc of the circle that reproduces the bundle shape obtained by the model to estimate the radius R and to calculate M_z from the exact solution. The estimate of the error on the twisting moment is equal to that of the bending moment because the exact solution for the helicity is a function of the ratio of the twisting moment to the bending moment (Fig. 4b).”

Reviewer #1: (minor points) The descriptions of the lengths of the scale bars are missing in Fig. 1, Fig. 3 and Fig. 4.

Authors: We added the descriptions of the scale bar lengths in captions for Fig. 1, Fig. 3 and Fig. 4.

Reviewer #3 (Remarks to the Author):

Reviewer #3 thanks authors for their response and manuscript revisions. My previous reference to concurrent work by Yu, Redemann et al was mistaken due to a search engine error.

The only point remaining is attribution of torsion to molecular motor torque. The mention in the authors' revision of alternative mechanisms is sufficient at this point to proceed with publication and allow further discussion in the wider scientific community since the experimental evidence in the paper should be disseminated. I however do not agree with the arguments put forward in the authors' Rebuttal letter.

Even within a linear model coupling between extension and torsion is quite possible, possibly strong, and well documented:

- "The torsion-extension coupling in pretwisted elastic beams", S. Krenk, International Journal of Solids and Structures

Volume 19, Issue 1, 1983, Pages 67-72

- "On Timoshenko-like modeling of initially curved and twisted composite beams", By:Yu, WB (Yu, WB); Hodges, DH (Hodges, DH); Volovoi, V (Volovoi, V); Cesnik, CES (Cesnik, CES)
INTERNATIONAL JOURNAL OF SOLIDS AND STRUCTURES

Volume: 39

Issue: 19

Pages: 5101-5121

Published: SEP 2002

- "Direct determination of DNA twist-stretch coupling"

By:Kamien, RD (Kamien, RD); Lubensky, TC (Lubensky, TC); Nelson, P (Nelson, P); OHern, CS (OHern, CS)

EUROPHYSICS LETTERS

Volume: 38

Issue: 3

Pages: 237-242

Published: APR 20 1997

The key hypothesis of a Kirchhoff rod that is not satisfied for a microtubule is that the cross-section of a segment remains undistorted and perpendicular to axis defined by third dimension (e.g., Cytoskeleton, 75(2):45-60, 2018). Eq. (5) is then no longer valid since longitudinal force directly produces torsion. The argument that inactivation of kinesin-5 did not lead to overall change of spindle length is an interesting conjecture, but not convincing since they are not the sole determinants of spindle length. The interesting prediction from the paper is the number of motors (10-100) per bundle needed to produce the twisting moment. This is worth further experimental study.

Authors: We added the reference Cytoskeleton, 75(2):45-60, 2018 after the following sentence in Discussion:

“Alternatively, linear forces acting on microtubules may lead to torsion due to a helical arrangement of tubulin subunits in the microtubule^{44,45}.”

Reviewer #3: Given the above, the claim in the Abstract that "bending and twisting moments generate curved shapes" seems too strong, and should be replaced by "may generate", similar to the more careful statement of conclusions in the revised Discussion section.

Authors: We softened the sentence in the Abstract:

“Our theoretical model predicts that bending and twisting moments may generate curved shapes of bundles.”